# Remote Sensing Image Target Detection and Recognition Based on YOLOv5

Xiaodong Liu, Wenyin Gong [ORCID], Lianlian Shang *, Xiang Li [ORCID] and Zixiang Gong

School of Computer Science, China University of Geosciences, Wuhan 430074, China;
liuxiaodong@cug.edu.cn (X.L.); wygong@cug.edu.cn (W.G.); lixiang@cug.edu.cn (X.L.); islvjh@cug.edu.cn (Z.G.)
* Correspondence: s1202240424@cug.edu.cn

**Abstract:** The main task of remote sensing image target detection is to locate and classify the targets of interest in remote sensing images, which plays an important role in intelligence investigation, disaster relief, industrial application, and other fields. However, the targets in the remote sensing image scene have special problems such as special perspective, scale diversity, multi-direction, small targets, and high background complexity. In this paper, the YOLOv5 target detection algorithm is improved according to the above characteristics. Aiming at the problem of large target size span in remote sensing images, this paper uses K-Means++ clustering algorithm to eliminate the problem in which the original clustering algorithm is sensitive to initial position, noise, and outliers, and optimizes the instability caused by K-Means clustering to obtain preset anchor frames. Aiming at the redundancy of background information around the location of remote sensing image targets, the large number of small targets, and the denseness of targets, a double IoU-aware decoupling head (DDH) is introduced at the output end to replace the coupled yolo head, which eliminates the interference caused by different task sharing parameters. At the same time, the correlation between positioning accuracy and classification accuracy is improved by the IoU-aware method. The attention mechanism is introduced into the backbone network to optimize the detection and small target detection in complex backgrounds. The mAP of the improved YOLOv5 algorithm is improved by 9%, and the detection effect of small targets and dense targets is significantly improved. At the same time, the paper has achieved good results through the verification of the DIOR remote sensing data set and has also achieved good performance advantages in comparison with other models.

**Keywords:** remote sensing image; object detection; double IoU-aware decoupled head; K-Means++; attention mechanism

## 1. Introduction

When performing target detection from remote sensing images, the most important thing is to extract the necessary information from the dataset. Traditional information extraction prior to deep learning target detection methods is a three-step process: image data pre-processing, image feature extraction, and feature processing. The most commonly used detection operators for traditional detection methods mainly include corner point detection operator, orientation gradient histogram [1], hal features [2], scale invariant feature transform [3], etc. However, remote sensing images are often of high resolution and have a large amount of image data, and still using traditional detection methods on this basis can result in a great waste of time and manpower.

In recent years, with the advancement of technology, deep learning algorithms have been proved to be a very excellent image processing technology. Their ability to automatically extract feature information in images greatly reduces labour costs. Especially for high-resolution and large-scale remote sensing image target detection, the use of deep learning technology can reduce the workload of traditional detection algorithms to manually design algorithms, thus greatly expanding the scope of application and having a high

detection effect. With the continuous improvement of neural networks and the widespread use of deep learning in image processing, the processing of remote sensing images has become more efficient and accurate. Especially in target detection, more and more researchers have begun to use this technology to develop more advanced deep convolutional neural networks to meet the changing environmental needs.

Collectively, current deep learning target detection frameworks are broadly divided into two types, one is the dual-stage detector, such as R-CNN [4], and the variants Fast-RCNN [5], Faster-RCNN [6], and Mask-RCNN [7] based on it. The other is the single-stage target detector, which does not have a separate candidate region generation network, but instead brings all processes together into a single branch that will treat all locations on the image as regions to be detected for target recognition and classification. The YOLO (You Only Look Once) [8] target detection framework, proposed by R. Joseph et al. in 2016, enables more efficient target detection by partitioning the input image into multiple grids (grid cells) by means of sliding windows. The difference between this algorithm and the two-stage detector is that it converts complex object detection tasks into simple regression tasks and does not require regional recommendations. Therefore, YOLO is the first single-stage object detector in the field of deep learning. If the centre point of the target is mapped to a grid, it can be used to predict the category of the target. In addition, each grid can be used to predict the regression parameters of the bounding box and a confidence value for object detection. The fast version detection speed of the framework can reach 155 fps, but the detection effect of small targets is not good, and the target overlapping environment is prone to missed detection. In 2020 Alexey Bochkovskiy proposed YOLOv4 and YOLOv5, further improved versions of YOLO series detectors, and the accuracy and speed of detection were improved. In 2020, Wu et al. proposed the IoU-aware algorithm. This algorithm sets a new confidence level as the product of the predicted IoU value and the classification score by improving the correlation between the classification accuracy and the positioning accuracy predicted by the single-stage detection model. The model improves AP by 1.7–1.9% on the COCO dataset and 2.9–4.4% on the PASCOL VOC. In 2022, Wang et al. proposed the Double IoU-aware Decoupled Head (DIDH) [9], which greatly improves the localization accuracy of the model, with an average accuracy improvement of 2.4% on the PASCOL VOC2007 dataset, and a convergence speed cycle of only a quarter of that of DETR. SSD is also widely used in single-stage networks, in addition to the YOLO family of algorithms [10].

Although many scholars have improved the target detection algorithm, the size span of the target in the remote sensing image is large. There are many types of targets in the image and the sizes between the targets are far apart. The performance requirements of the detection network are high, and the general detection network is difficult to directly detect remote sensing images. Because the remote sensing images taken by overlooking usually contain a large area of terrain background, this redundant background information brings some difficulties to the network in detecting targets. At the same time, because there are very small targets in the image taken far away from the target, these targets are more difficult to detect, resulting in an increase in the probability of missed detection and false detection.

Therefore, in the case of studying the relevant principles and development status of the current deep neural network target detection algorithm, combined with the characteristics of remote sensing images, this paper explores the structure and technical improvement of the target detection algorithm in the remote sensing image scene to improve the performance of the target detection algorithm of the neural network in the remote sensing image scene. This paper studies the improvement scheme of the target detection algorithm based on YOLOv5. The existing problems and improvement schemes are as follows.

(1)　The introduction of K-Means++ clustering in the input stage to generate anchor frames can improve the drawbacks of the K-Means clustering algorithm, effectively suppress the influence of noise points and outliers on the clustering of anchor frames, so as

to obtain more accurate results, and can reduce the difficulty of model convergence, thus better meeting the needs of remote sensing target detection.

(2) The attention mechanism is similar to the processing of information by the human brain, which is mainly derived from the study of human vision. Humans will accept all information acquired by the eyes without thinking, but the human brain will often automatically filter out irrelevant information that is not relevant to the task at hand, and focus cognitive attention on the main information, so as to make rational use of the limited visual information. Therefore, the attention mechanism was added to YOLOv5's DarkNet53 backbone network to achieve the model's improved accuracy for target recognition of small targets, in complex background scenes, and in mutual occlusion.

(3) The original YOLOv5 used a coupled yolo head for predicting confidence, classification scores, and localisation regression results. However, the feature information focused on the localisation task and the classification task are different, and the coupled detection head, by sharing the convolution kernel and thus sharing parameters, will in turn affect the final results between the different tasks. At the same time, the separation of the localisation and classification tasks will lead to a mismatch between the final localisation accuracy and the classification accuracy, thus compromising the final accuracy of the model. The improvement solution uses the introduction of the double IoU-aware decoupled head to decouple the detection head classification task from the regression task into two branches, while redesigning the confidence level to improve the correlation between classification accuracy and localization accuracy, thus improving the detection effect and convergence speed of the model.

## 2. Methods

### 2.1. YOLOv5 Target Detection Algorithm

The overall structure of the YOLOv5 target detection algorithm can be broadly divided into four parts: the input, backbone, Neck network, and the output Head. The Neck including SPPF and New CSP-PAN, and the Head still using the YOLOv3 Head.

Focus structure is a feature extraction module that focuses the model's attention on regions with important information by adjusting the resolution of the input feature map before the image enters the backbone network, with the main processes going through scale reduction, channel blending, scale amplification, and channel blending. However, in the actual YOLOv5 code, a $6 \times 6$ sized convolutional layer is used to replace the Focus module prior to the backbone network, and the two are equivalent in theoretical effect, with the above changes being more efficient for using GPU devices.

The YOLOv5 target detection network uses the CSP module in backbone, which reduces the computational effort of the network during forward inference. The CSP module consists of the conv layer, the BN layer, and the SiLU activation function.

YOLOv5 has made some significant changes to the Neck. Firstly, the SPP part was replaced by the SPPF designed by Glenn Jocher, which has the same function but is more efficient. The SPP structure takes the input feature maps and passes them through multiple MaxPools of different sizes in parallel.

The SPPF structure takes the input feature matrix and feeds the backbone into multiple $5 \times 5$ MaxPool layers in a serial manner. The SPPF can therefore significantly reduce the number and thus, SPPF can significantly reduce the computational effort and running time associated with the SPP network structure. Similarly, three $5 \times 5$ MaxPool layers are equivalent to one $13 \times 13$ MaxPool layer. Ultimately, by stitching the serial one to three MaxPool layers together with the original input feature matrix through Concat, the same effect can be achieved with the SPP structure, and the serial $5 \times 5$ size MaxPool is more efficient than the $9 \times 9$ and $13 \times 13$ size MaxPool, so the use of the SPPF structure in the network structure improves the efficiency of the network using the SPP structure.

The results are the same for both comparisons, and the analysis from the table shows that the SPPF module executes in less than half the time of the SPP module, and runs more than twice as efficiently, thereby showing the optimised performance of the SPPF module.

The total loss function of the YOLOv5 model is defined as the sum of three types of losses as shown in Equation (1).

$$Loss = \lambda_1 L_{cls} + \lambda_2 L_{obj} + \lambda_3 L_{loc} \tag{1}$$

In YOLOv5, both the confidence loss and the target category loss are calculated using binary cross-entropy loss [11].

In YOLOv5, a balancing of the different scales of target loss is performed simultaneously, and the different scales are used to apply different weights to the obj loss on the three different prediction feature layers. The formula for balancing the different scales of loss is shown in Equation (2), where the coefficients are the default hyperparameters.

$$L_{obj} = 4.0 * L_{obj}^{small} + 1.0 * L_{obj}^{medium} + 0.4 * L_{obj}^{large} \tag{2}$$

The YOLOv5 model for overlapping area uses CIoU [12] as the loss function for the bounding box regression, the CIoU method overcomes the disadvantages of IoU while making full use of the advantages of IoU while combining the centroid Euclidean distance between the prediction frame and GTBox and the diagonal distance between the two bounding boxes, where $\alpha$ is the penalty factor. the formula for CIoU is as in Equation (3) and the loss function CIoUloss is calculated as in Equation (4).

$$CIoU = IoU - \frac{\rho^2\left(b, b^{gt}\right)}{c^2} - \alpha v \tag{3}$$

$$CIoULoss = 1 - CIoU \tag{4}$$

### 2.2. YOLOv5 Input Stage Improvements

The anchor box generation method in the input stage of YOLOv5 is still the method of generating a priori box by continuing YOLOv3's K-Means clustering. The default template is also the natural image target template obtained on the coco dataset, which is not applicable to remote sensing scene targets. The K-Means algorithm is very sensitive to noise and outliers, which can lead to inaccurate cluster division, and is also sensitive to the initial cluster centre. The target size span in remote sensing images is large, and the number of targets in different categories is also different. The effect of the K-Means clustering algorithm will be more affected.

In response to the shortcomings of K-Means, such as noise and outliers, K-Means++ [13] was introduced to optimise the original generation of unstable prior frames. The K-Means++ algorithm is an improvement on the traditional K-Means clustering algorithm and aims to improve the selection of initial clustering centres to improve the convergence speed and clustering quality of the algorithm. The formula for this is Equation (5). A new centre is obtained mainly by randomly selecting an initialised cluster centre and calculating the distance between each data point and the centre, based on the probability obtained from the calculation. Where the shortest distance is noted as D(x) and the probability of the resulting sample centre is P(x).

$$P(x) = \frac{D(x)^2}{\sum_{x \in X} D(x)^2} \tag{5}$$

The above process was repeated until K clustering centres were selected. Three detection scales were set for the DOTA dataset and the output structure of YOLOv5, and three standard anchor boxes were generated for each detection scale. The anchor box

benchmarks for the anchor boxes generated using the K-Means++ clustering algorithm are shown in Table 1.

**Table 1.** Corresponding anchor frame benchmarks at different scales.

| Output Layer | Size | Size | Size |
|:---:|:---:|:---:|:---:|
| P3 | [11, 11] | [14, 23] | [26, 13] |
| P4 | [24, 25] | [26, 47] | [47, 31] |
| P5 | [54, 55] | [94, 115] | [190, 192] |

*2.3. Backbone Network Improvements*

Although the backbone of YOLOv5 adopts the advantages of New darknet53 with strong learning ability and low computational complexity, the direct use of the original YOLOv5 algorithm in remote sensing image target detection is not ideal, especially for small targets. There are two main reasons.

(1) In the backbone network stage of YOLOv5, a large number of convolution operations are used to make the pixels of small targets in the feature map smaller and smaller, and the feature information containing small targets is less and less. For image feature information, multi-layer convolution will lose a lot of feature information, which will affect the detection effect of small targets.

(2) In the remote sensing image, the proportion of the target pixel relative to the entire image is very small, and there is complex background information. However, the YOLOv5 algorithm does not focus on important information, and cannot distinguish irrelevant background information from noise and important information, which leads to the incomplete and insufficient extraction of the feature information of the remote sensing target by the detection model.

The attention mechanism of neural networks allows the limited computational resources to be focused on more important tasks, given the limited computational power. Typically, the more parameters a neural network model has, the more expressive the model is and the more features it can learn, but at the same time this brings with it the problem of information overload. In neural networks, the attention mechanism allows the model to weight the input information, allocating a higher level of attention to information relevant to the task at hand. Through learning, the attention mechanism can automatically decide which input information is most important for the current task. This mechanism allows the neural network to focus on the critical parts of the vast amount of input information and ignore those parts that are not important for the current task. Therefore, introducing the attention mechanism to improve the YOLOv5 algorithm and applying it to remote sensing image target detection can better scan the whole remote sensing image and exclude irrelevant background information interference.

(1) Principle of SENet

SENet [14] was proposed in 2017 as a deep neural network structure for image classification tasks, which introduces an attention mechanism that adaptively learns to correlations between different channels in the feature map. The specific structure is shown in Figure 1.

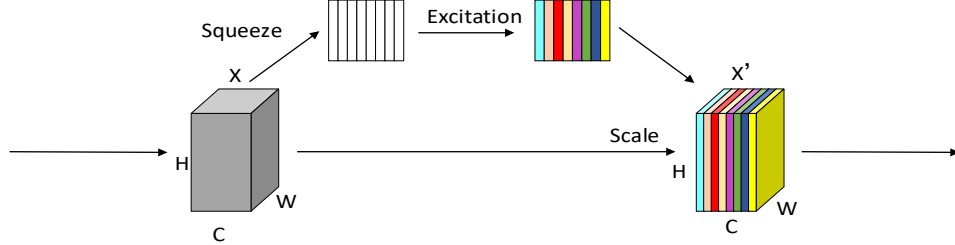

**Figure 1.** SENet structure.

The SENet module structure diagram shows that SENet achieves this by introducing a structure called the Squeeze-and-Excitation block, which consists of two main steps: Squeeze and Excitation. In the Squeeze phase, SENet compresses each channel's feature map into a feature vector by means of a global average pooling operation to obtain the weights on the channels. This feature vector reflects the extent to which each channel contributes to the overall feature map. The formula is expressed as Equation (6).

$$Z_c = F_{eq}(u_c) = \frac{1}{H \times W} \sum_{i=1}^{H} \sum_{j=1}^{W} u_c(i,j) \tag{6}$$

In the Excitation phase, SENet uses a fully connected layer to learn a weight vector that corresponds to the importance weights of each channel. By multiplying this weight vector with the original feature map on a channel-by-channel basis, SENet can enhance useful features and suppress useless ones. The formula is shown in Equation (7).

$$S = F_{ex}(z, W) = \sigma(g(z, W)) = \sigma(W_2 \delta(W_1 z)) \tag{7}$$

(2)　CBAM

The CBAM (convolutional block attention module) [15] is an attention mechanism for a variety of tasks such as image classification, target detection, etc., to enhance feature representation in deep neural networks. The structure of the CBAM attention mechanism is shown in Figure 2.

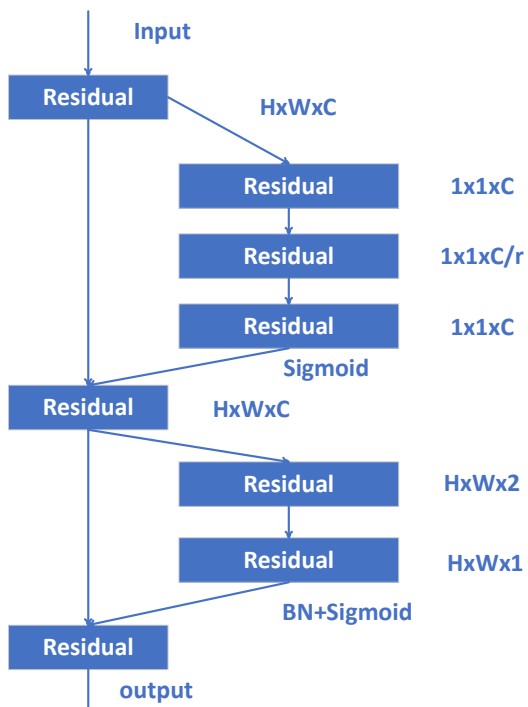

**Figure 2.** Structure of the CBAM attention mechanism.

As can be seen from the CBAM structure diagram, the module structure consists of two main sub-modules, including the channel attention mechanism module similar to the channel attention and spatial attention mechanism modules in SENet. The channel attention module enhances the useful channels in the feature map by learning the importance of each channel. It obtains a global feature description for each channel through a global averaging pooling operation and learns the weights of the channels through two fully connected layers. These weights are then used to weight the summation of the feature maps for each

channel to enhance the useful feature channels and suppress the useless ones, by increasing the weight distribution over the channels on the original feature maps, as in (8):

$$M_c(F) = \sigma(MLP(AvgPool(F) + MLP(MaxPool(F)))) \tag{8}$$

The Spatial Attention Module enhances the spatial information of a feature map by learning its importance at different spatial locations. It captures the importance of each channel using channel maxima and averages, and then generates a spatial attention map through a convolution operation. This attention map is used to perform a channel-by-channel weighting operation on the original feature map to enhance the important spatial locations and suppress the unimportant ones. The formula for this module is (9):

$$M_s = \sigma\left(f^{7\times7}([AvgPool(F); MaxPool(F)])\right) \tag{9}$$

By combining channel attention and spatial attention, the CBAM module can adaptively learn the relationship between channel relevance and spatial importance in the feature map, thus improving the feature representation capability.

(3) CA

The CA attention mechanism [16] module is an attention mechanism for computer vision tasks designed to enhance the channel relevance of feature maps in convolutional neural networks (CNNs). The CA attention mechanism adaptively adjusts the contribution of channel features by learning the importance weight of each channel, thereby improving the capability of feature representation. The implementation of the CA attention mechanism is shown in Figure 3.

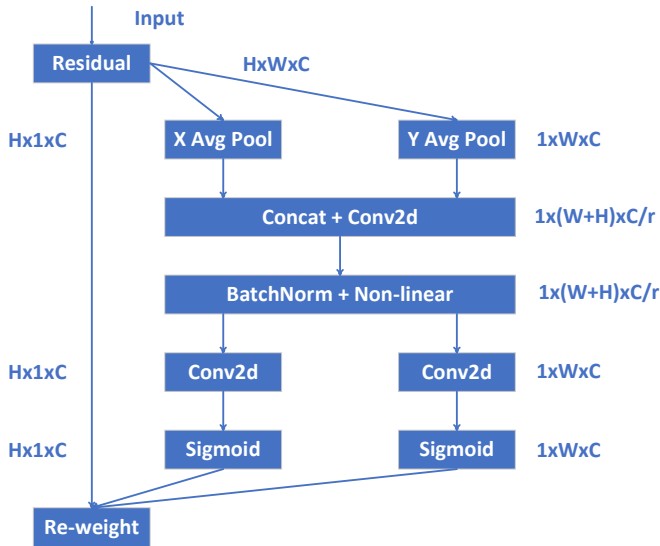

**Figure 3.** Structure of CA attention mechanism.

The CA attention mechanism encodes the channel relationship and long-term dependence by accurately positioning information. The specific operation is divided into two steps: information embedding and attention generation. The calculation formulae are shown in Equations (10) and (11).

$$z_c^h(h) = \frac{1}{W}\sum_{0 \le i \le W} x_c(h, i) \tag{10}$$

$$z_c^w(w) = \frac{1}{H}\sum_{0 \le j \le H} x_c(j, w) \tag{11}$$

The main idea of the CA attention mechanism is to use global information to capture the importance of each channel. It does this by introducing two steps: 1. Squeeze: the features of each channel in the feature graph are compressed through a global average pooling operation to obtain a global description vector for each channel. This global description vector reflects the importance of each channel to the overall feature map. 2. Excitation: a small multilayer perceptron (MLP) network is used to nonlinearly map and excite the global description vector. This MLP network usually consists of one or more fully connected layers for learning the weights of the channels. Through this process, the CA attention mechanism is able to learn the importance weights of each channel to enhance useful feature channels and suppress useless ones.

In the experiments, the attention mechanism is added to the C3 module of the YOLOv5 backbone network (Figure 4 shows where the attention mechanism is inserted) and the weights of the attention mechanism are rescaled once before the final output of the C3 module to improve the performance of the target detection algorithm for interfering information and small targets.

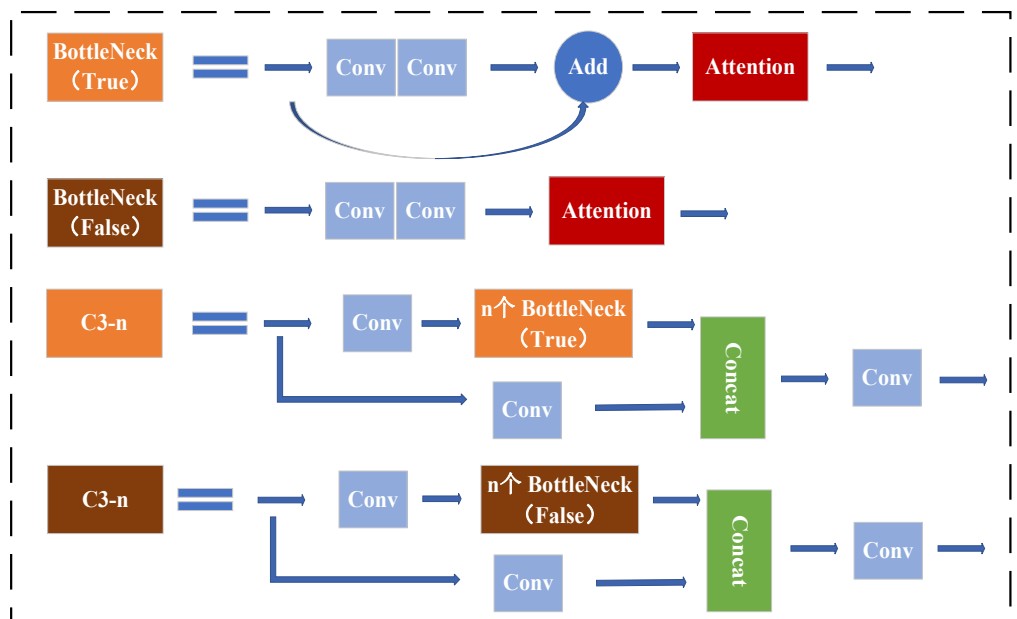

**Figure 4.** Location of attention mechanism additions.

### 2.4. Output Improvements

The output of YOLOv5 still uses a yolo head. The original yolo head uses the same convolutional kernel for the localisation task and the classification task, sharing the same parameters, but the information features of the two tasks are not the same, and the coupled detection head can greatly interfere with the accuracy of the two tasks. The classification task mainly learns features that focus on the locally representative feature information of the category, while the localisation task focuses on the contour information of the target. Coupling the detection side of the two subclass tasks even though the two subclass tasks focus on different image features, back-propagating the tuning parameters during training, and sharing the learned parameters will greatly affect the detection accuracy of the two subclass tasks.

The decoupled detection head is proposed in the YOLOX target detection algorithm, as shown in Figure 5 The separation of the two types of tasks into a classification sub-network and a localisation sub-network eliminates the sharing of parameters when coupled, allowing both task sub-networks to learn to focus only on the relevant tasks and improve detection results.

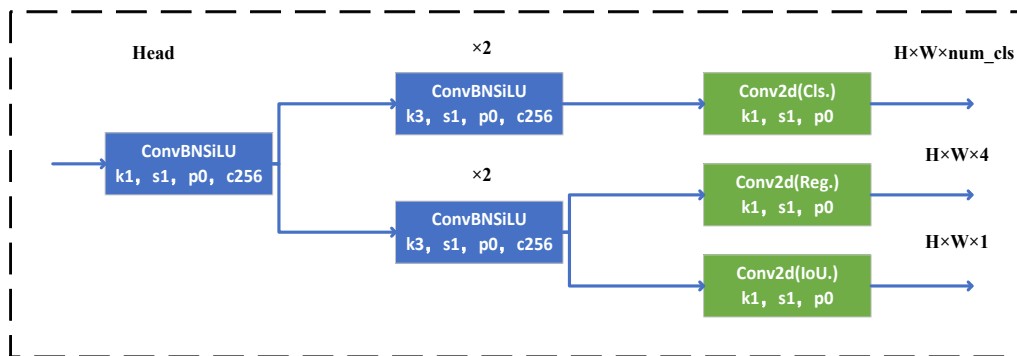

**Figure 5.** Structure of the YOLOX decoupling head.

However, since the classification and localisation sub-networks are trained from independent objective functions that are not previously correlated with each other, the correlation between the classification score and localisation accuracy is low, which will seriously affect the localisation accuracy of the model. It was also demonstrated in the paper IoU-aware RetinaNet [17] that independent classification and localisation sub-networks predict the classification score of each anchor without knowing the localisation accuracy (IoU score) after the model converges, and that there is a mismatch between classification accuracy and localisation accuracy, such as a high classification score but low IoU, versus a low classification score but high IoU; these detections can compromise the average model accuracy in both directions.

To address this problem, the paper introduces DDH (Double IoU-aware Decoupled Head) [6] at the output side to solve both the coupling problem and the sub-network correlation problem, and the structure diagram of DDH is shown in Figure 6.

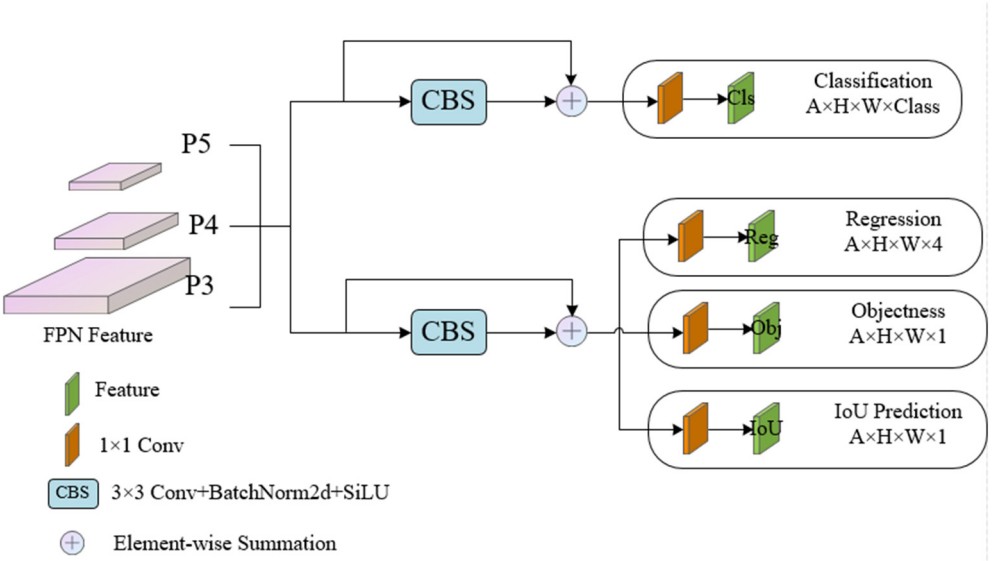

**Figure 6.** DDH structure diagram (Quoted from Reference [6]).

By decoupling the yolo head and introducing the residual edge structure of the CBS (conv + batch normalization + SiLU) module in each branch sub-network, DDH can effectively improve the detection effect and at the same time can improve the convergence speed to achieve a better convergence effect. The IoU-aware loss is calculated using the binary cross-entropy loss denoted as $L_I$ in Equation (12), and $\widehat{IoU_i}$ is calculated as Equation (13).

$$L_I = \frac{1}{N_{Pos}} \sum_{i \in Pos}^{N} BCE(IoU_i, \widehat{IoU_i}) \tag{12}$$

$$\widehat{IoU_i} = overlap\left(b_i, \widehat{b}_i\right) \tag{13}$$

The total loss after using DDH is calculated as Equation (14); $L_o$ is the confidence loss, $L_c$ is the classification loss, $L_r$ is the regression loss, and $W_c$ and $W_o$ are the weighting coefficients of the corresponding losses, respectively.

$$L = W_c L_c + W_o L_o + W_r(L_r + W_I * L_I) \tag{14}$$

The new confidence level set for improving the correlation between classification accuracy and localization accuracy is the product of the final obtained classification score with the original confidence score and IoU value, where $\alpha$ is the value between [0, 1] used to control the contribution of the object score and IoU score to the final confidence level, as shown in Equation (15).

$$S_{det} = Cls_i * Obj_i^\alpha * IoU_i^{1-\alpha} \tag{15}$$

This confidence level can perceive both the classification accuracy and the degree of overlap between the predicted borders and the real borders, enabling the selection of samples with both high localization and classification scores for learning, thus improving the detection effect of the model.

In summary, based on the differences between targets in remotely sensed images and natural images, such as the fact that targets in remotely sensed images often appear in complex background environments, that the size of targets in remotely sensed images is richer in texture information, and that the resolution of remotely sensed images is greater and often contains more targets with a large span of sizes between different targets, three stages of adaptation are made to the YOLOv5 detection model for remote sensing scenes. The final improved YOLOv5 detection model has been adapted to the remote sensing scenario. The structure diagram of the final improved YOLOv5 model is shown in Figure 7.

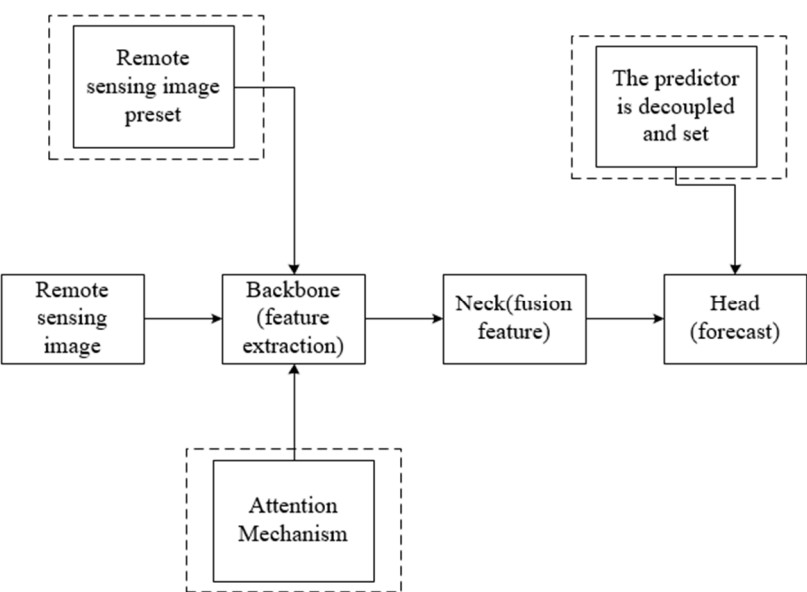

**Figure 7.** Overall structure of the improved YOLOv5.

## 3. Experiment and Result Analysis

### 3.1. Experimental Setup

In this section, using the DOTA dataset, ablation experiments were conducted on each of the three improvement stages of the YOLOv5 detection model to verify the improvement of the original model. After the ablation experiment, the optimal model was validated using the DIOR dataset and further validated with other current detection models.

### 3.1.1. Dataset

The remote sensing image dataset used in the thesis experiments is from the DOTA [18] dataset. We collected 2806 aerial images from different sensors and platforms in the DOTA dataset; each image is approximately 4000 × 4000 pixels in size and contains objects of various scales, orientations, and shapes. The training set contained 1411 images, the validation set contained 458 images, and test set contained 937 images, with the main sources of images being Google Earth 6.0, the JL-1 satellite, and the GF-2 satellite. Fifteen categories were included, namely aircraft, ships, oil tanks, baseball fields, tennis courts, basketball courts, athletic fields, harbours, bridges, large vehicles, small vehicles, helicopters, roundabouts, football fields, and swimming pools.

As the size fraction of remote sensing images is too large, the images are pre-processed by cropping them to a pixel size of 800 × 800 and setting a threshold of 0.7 for the target size left after cropping; values less than 0.7 marked them as negative samples. The statistics of the number of images in the training set of the DOTA dataset after pre-processing are shown in Figure 8, and the statistics of the number of targets in each category are shown in Figure 9.

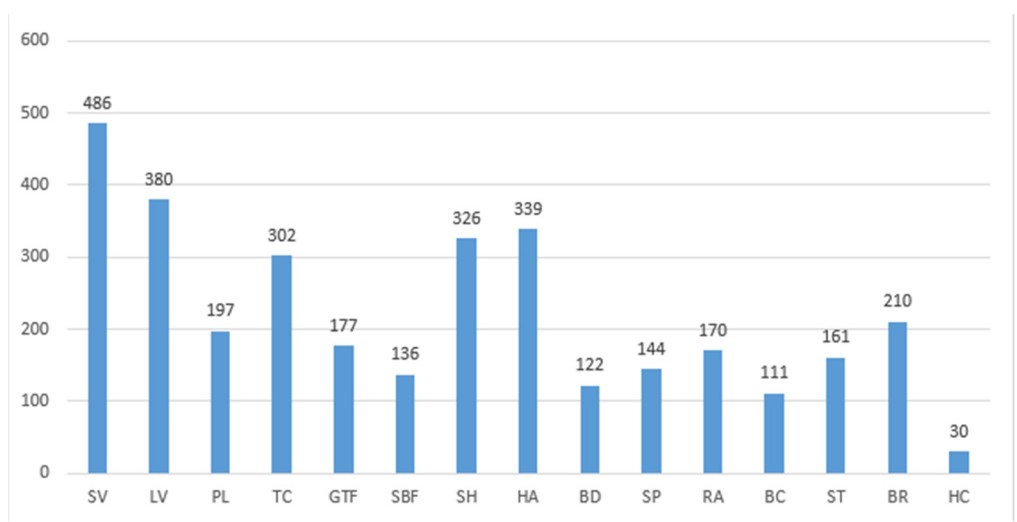

**Figure 8.** DOTA data set training set picture number statistics.

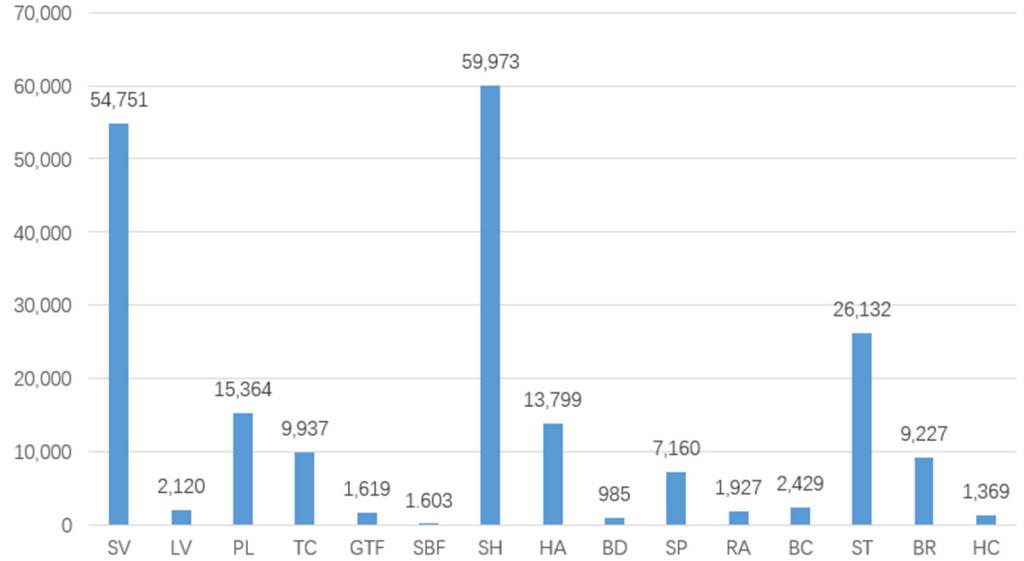

**Figure 9.** Statistics on the number of targets in each category in the DOTA dataset.

### 3.1.2. Evaluating Indicator

(1) Accuracy

The accuracy rate is defined as the proportion of the predicted correct results in the total sample, which is expressed as Formula (16).

$$\text{Accuracy} = \frac{TP + TN}{TP + TN + FP + FN} \tag{16}$$

Although the accuracy seems to be very intuitive, it is not scientific to evaluate the model only by accuracy. Algorithms with high accuracy are not necessarily good. Moreover, the accuracy evaluation index is not applicable in the classification task of category imbalance, and it may even be misleading. Therefore, the paper adopts a new evaluation index.

(2) TP, FP, TN, and FN

TP (true positive) is the real example, which refers to the number of positive examples being classified by the model. TN (true negative) refers to the number of negative examples correctly classified by the model; fP (false positive) is a false positive, which means that the model incorrectly classifies the counterexamples into the number of positive examples, that is, multiple measurements; fN (false negative) indicates the model incorrectly classifies positive examples into the number of counterexamples, that is, missed measurements.

(3) Recall rate and precision rate

Recall is the recall rate, which refers to the ratio of the number of positive cases correctly identified by the model to the actual number of positive cases. The calculation method is shown in Formula (17).

$$\text{Recall} = \frac{TP}{TP + FN} \tag{17}$$

Precision represents the ratio of the number of positive examples correctly identified by the model to the number of positive examples predicted by the classifier. The formula is shown in (18).

$$\text{Precision} = \frac{TP}{TP + FP} \tag{18}$$

(4) AP and mAP

AP (average precision) is an indicator used to measure the balance between the precision rate and the recall rate of the target detection algorithm on a single category, that is, the average of the precision on the P–R curve, expressed as Formula (19).

$$\text{AP} = \sum_{i=1}^{n} P(i)\Delta r(i) = \int_{0}^{1} p(r)dr \tag{19}$$

The mAP (mean average precision) is obtained by averaging the APs of multiple categories of targets, which is used to evaluate the performance of the detection model on the entire data set. The calculation formula is shown in Formula (20).

$$\text{mAP} = \frac{\sum_{n=1}^{N} AP(n)}{N} \tag{20}$$

The n in the formula represents the category, and N represents the total number of categories.

(5) FPS

FPS (frames per second) is a commonly used evaluation index in deep learning to measure the processing speed of an algorithm or system. It represents the number of frames processed or displayed per second and is usually used to evaluate the real-time performance of an algorithm or system.

### 3.1.3. Experimental Environment

The experimental environment for this experiment was Windows 10, the hardware included an 11th generation i5-11400 processor and 16 G of RAM, PyTorch version 1.8, and was equipped with an NVDIA RTX2060 graphics card with 8 GB of video memory, the configuration of the experimental environment for subsequent chapters remained unchanged.

### 3.1.4. Experimental Parameters

YOLOv5 improved the training of the experiments by setting the batch_size to 8 and randomly selecting eight samples of data from the training set each time for a total of 200 epochs of training. Table 2 shows the setting of hyperparameters in the experiment.

**Table 2.** Hyperparameter configuration.

| Initial Learning Rate | Recurrent Learning Rate | Learning Rate Momentum | Weight Decay Factor | Warm-Up Learning Rounds | Warm-Up Learning Momentum | Warm-Up Initial Learning Rate |
|---|---|---|---|---|---|---|
| 0.01 | 0.2 | 0.937 | 0.0005 | 3.0 | 0.8 | 0.1 |

The baseline model for this experiment is the YOLOv5m model, and the convergence of the three stages of the improved YOLOv5 model during training is shown in Figure 10. From the figure, we can see that the improved network model had basically dropped to below 0.05 for all types of losses after about 50 epochs, while the learning rate further decreased after 50 epochs, and the loss value continued to slowly decrease as the training progresses, and the network model basically reached convergence after 100 epochs. The category loss, localization loss, and obj loss of the training and validation sets converged normally. The convergence of all losses was observed, and the model was stable in the pre-defined epochs, meeting the expectation before the experiment. The same training strategy will be maintained in the subsequent experiments, and the relevant hyperparameters will be set to the same level.

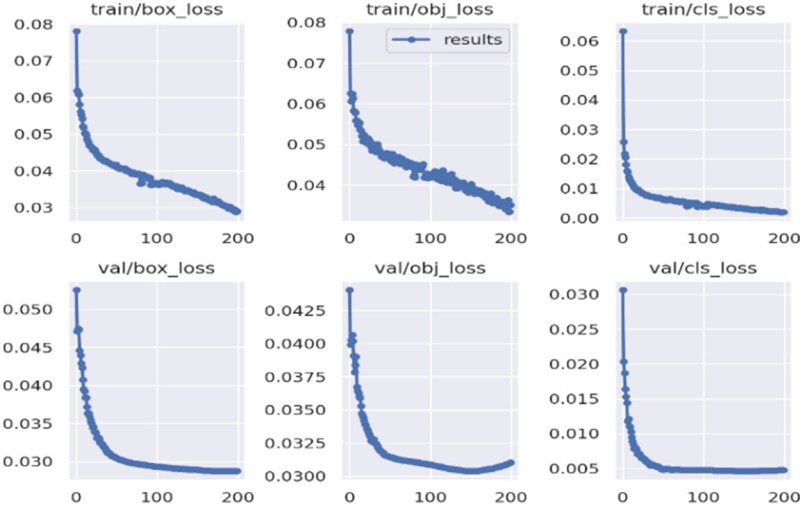

**Figure 10.** Network training convergence.

### 3.2. Analysis and Comparison of Results

(1) Input stage improvement

The anchors templates obtained from the K-Means++ clustering in Section 2.2 were applied to the subsequent experiments. The anchors templates are shown in Table 3, and P3,

P4, and P5 are the output predicted feature layers, where the improved clustering method achieved a recall of 0.9977.

**Table 3.** Improved anchors template.

| Output Layer | Size | Size | Size |
|:---:|:---:|:---:|:---:|
| P3 | [11, 11] | [14, 23] | [26, 13] |
| P4 | [24, 25] | [26, 47] | [47, 31] |
| P5 | [54, 55] | [94, 115] | [190, 192] |

Table 4 compares the anchors template obtained from the original YOLOv5 using K-Means clustering on the coco dataset; however, the template targets the natural image targets and has some limitations for targets under remotely sensed images. The a priori frame template obtained by applying the K-Means++ clustering algorithm to the horizontally annotated DOTA dataset of remote sensing images improved by about 0.6% on mAP. The K-Means++ algorithm selected more suitable prediction frames and improved more on small targets and targets that appeared more frequently in the DOTA dataset, such as small vehicles, boats, large vehicles, and ports. The improvement was achieved for small targets and targets with a high frequency in the DOTA dataset, such as small vehicles, boats, large vehicles, and ports, but there was a slight decrease for other targets. The templates generated by K-Means++ will be applied in all subsequent experiments.

**Table 4.** Comparison of anchors template improvements.

| Models | YOLOv5 | YOLOv5 + K-Means++ |
|:---:|:---:|:---:|
| plane | 85.2 | **86.0** |
| baseball-diamond | 75.2 | **75.3** |
| bridge | 42.1 | **43.0** |
| ground-track-field | **57.2** | 56.8 |
| small-vehicle | 53.2 | **54.3** |
| large-vehicle | 75.2 | **78.1** |
| ship | 79.3 | **81.0** |
| tennis-court | **89.3** | 87.7 |
| basketball-court | **56.6** | 55.8 |
| storage-tank | **66.3** | 64.9 |
| soccer-ball-field | **50.3** | 47.1 |
| roundabout | 54.1 | **55.8** |
| harbour | 72.6 | **73.3** |
| swimming-pool | 53.3 | **60.2** |
| helicopter | **52.2** | 44.7 |
| mAP@0.5 | 64.2 | **64.8** |

(2) Output-side improvement comparison experiments

The experimental sequence was followed by the validation of the output-side improvement effect and then the optimization effect of the attention mechanism on the detection model. The output-side improvements were mainly applied to the DDH-improved coupled yolo head, which was validated on the DOTA dataset, as shown in Table 5. Therefore, it can be seen that the YOLOv5 model, combined with the DDH detection head, separated the coupling of shared parameters between the localization and classification tasks, while the introduction of the IoU-aware method increased the correlation between the model localization accuracy and classification accuracy, which integrally improved the model convergence speed and effect, enhanced the overall model detection performance, and improved the detection accuracy.

**Table 5.** Comparison experiments with improvements at the output.

| Models | YOLOv5 | Ours (YOLOv5 + DDH) |
|---|---|---|
| plane | 85.2 | **91.0** |
| baseball-diamond | 75.2 | **71.2** |
| bridge | 42.1 | **44.9** |
| ground-track-field | 57.2 | **57.7** |
| small-vehicle | 53.2 | **57.1** |
| large-vehicle | 75.2 | **82.6** |
| ship | 79.3 | **83.1** |
| tennis-court | 89.3 | **92.4** |
| basketball-court | 56.6 | **59.9** |
| storage-tank | 66.3 | **67.9** |
| soccer-ball-field | **50.3** | 47.1 |
| roundabout | 54.1 | **55.8** |
| harbour | 72.6 | **82.0** |
| swimming-pool | 53.3 | **60.2** |
| helicopter | **52.2** | 38.2 |
| mAP@0.5 | 64.2 | **67.6** |

(3)    Backbone network improvement comparison experiments

After the verification of the first two stages of improvement experiments, four sets of comparison experiments were set up on the basis of the above-mentioned experiments for the addition of the three attention mechanisms, including the first two stages of improvement without the addition of the attention mechanism model (called the first two stages of the model and the comparison experiments with the addition of the three attention mechanisms model). The analysis of the results shows that the first two-stage model combined with the SENet attention mechanism resulted in a 1.8% improvement in mAP on the DOTA dataset; the first two-stage model combined with the CBAM attention mechanism resulted in a 5.6% improvement in mAP on the DOTA dataset; and the method combined with the CA attention mechanism resulted in a 2.6% improvement in mAP. In summary, it can be concluded that the attention mechanism optimised the recognition effect of the algorithm on remote sensing targets, and that the CBAM attention mechanism was applied more effectively than the other two attention mechanisms.

From Table 6, we can also observe that the detection of small targets such as small cars, boats and bridges was improved significantly after the introduction of the attention mechanism, and the recognition accuracy of targets that are easily aggregated such as large vehicles and tennis courts was also improved to a certain extent after the introduction of the attention mechanism, but not as much as small targets such as small cars and ports, further verifying the superiority of the improved algorithm. The three attention mechanisms have similar detection results for several types of multi-scale, large-area mutually occluded targets such as ships and tennis courts.

**Table 6.** Comparison experiments of attentional mechanism improvements.

| Models<br>Attention Mechanisms | YOLOv5 + DDH<br>No | SENet | YOLOv5 + DDH<br>CBAM | CA |
|---|---|---|---|---|
| plane | 91.0 | 90.9 | **93.4** | 91.3 |
| baseball-diamond | 71.2 | 75.8 | 78.2 | 78.7 |
| bridge | 44.9 | 46.1 | **54.9** | 47.6 |
| ground-track-field | 57.7 | 64.8 | **68.2** | 67.5 |
| small-vehicle | 57.1 | 58.2 | **69.6** | 59.2 |
| large-vehicle | 82.6 | 80.4 | **86.0** | 83.2 |
| ship | 83.1 | 84.3 | 88.7 | 85.7 |
| tennis-court | **92.4** | 92.1 | 91.2 | 89.2 |

**Table 6.** *Cont.*

| Models Attention Mechanisms | YOLOv5 + DDH No | SENet | YOLOv5 + DDH CBAM | CA |
|---|---|---|---|---|
| basketball-court | 59.9 | 66.3 | **71.8** | 67.1 |
| storage-tank | 67.9 | 65.3 | **74.2** | 68.9 |
| soccer-ball-field | 47.1 | 54.6 | 55.5 | **55.6** |
| roundabout | 55.8 | 58.0 | **64.0** | 57.6 |
| harbour | 82.0 | 73.8 | **85.8** | 74.9 |
| swimming-pool | 60.2 | 60.7 | **61.0** | 58.6 |
| helicopter | 38.2 | 55.4 | 57.2 | **57.6** |
| mAP@0.5 | 67.6 | 69.3 | **73.2** | 71.1 |

Figure 11 shows the P–R curve of the most improved "YOLOv5 + DDH + CBAM", subsequently referred to as the improved YOLOv5m method in this paper. The graphical analysis shows that when the recall reaches 0.6, the accuracy of the model is still above 0.8. When the recall reaches 0.8, the model accuracy is still around 0.4 and several of the categories such as aircraft, large cars, and baseball stadiums are still accurate at 0.8 and above when the recall reaches 0.8. The AP for each category is the area enclosed by the P-R curve corresponding to its category and the coordinate axis. The large area enclosed by the entire P-R curve, and the fact that the balance point is close to the upper right corner, indicates that the model as a whole has excellent detection performance on the dataset, proving that its improvement is good and meets the expected standard.

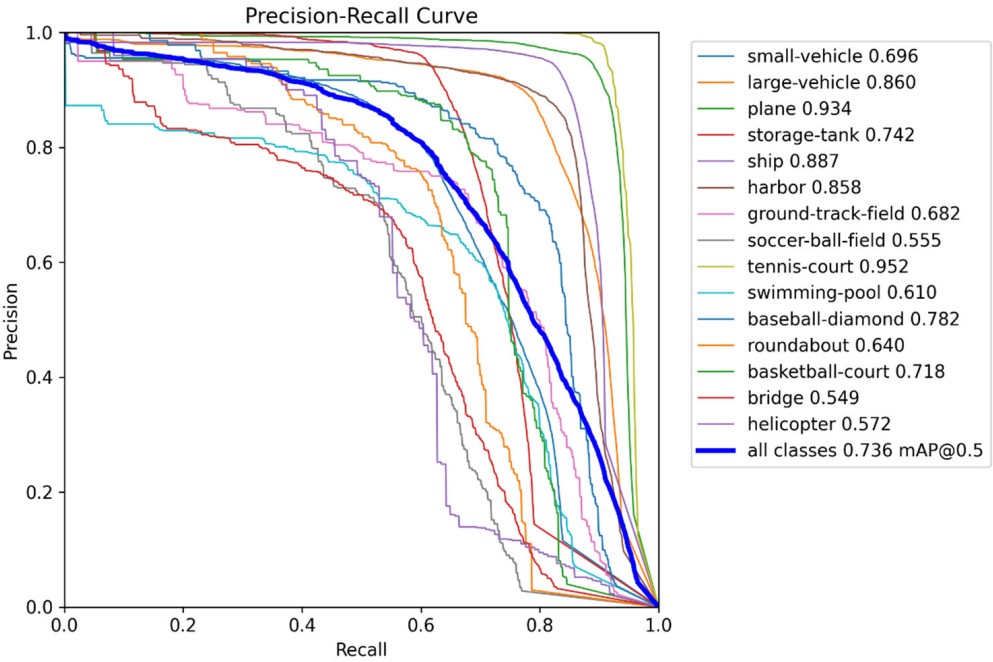

**Figure 11.** Improved P–R curve during YOLOv5 training.

Figure 12 shows the information related to the category labels in the dataset during training. In the figure, when training, the amount of data in each category of the data set is large, the number of small targets such as small vehicles and ships is large, and the samples are relatively unbalanced.

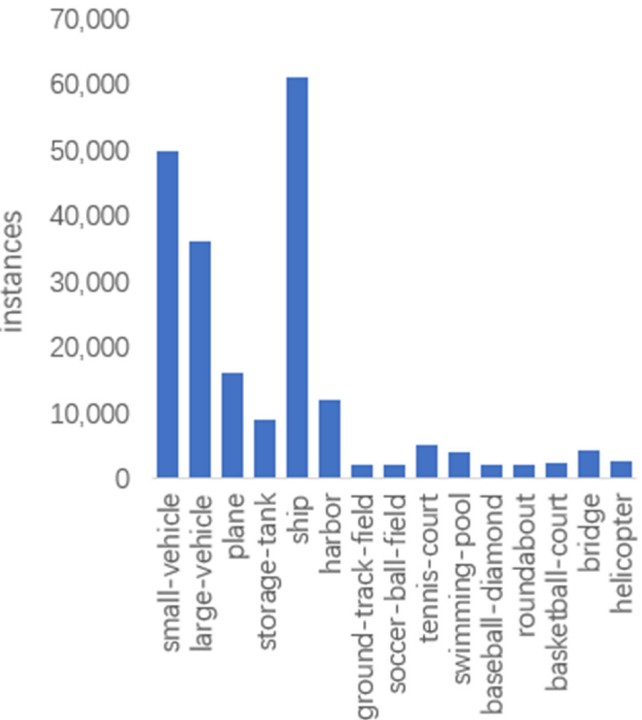

**Figure 12.** Related information such as category labels in the data set during training.

Figure 13 shows the F1 curve of the model. In the DOTA dataset with a more unbalanced sample, the majority of categories have high F1 scores, and some of the smaller number of categories can reach a maximum F1 score of 0.6 or higher, indicating that the model had a better effect on accuracy and recall. The improvement is a further improvement in the overall effectiveness of the model.

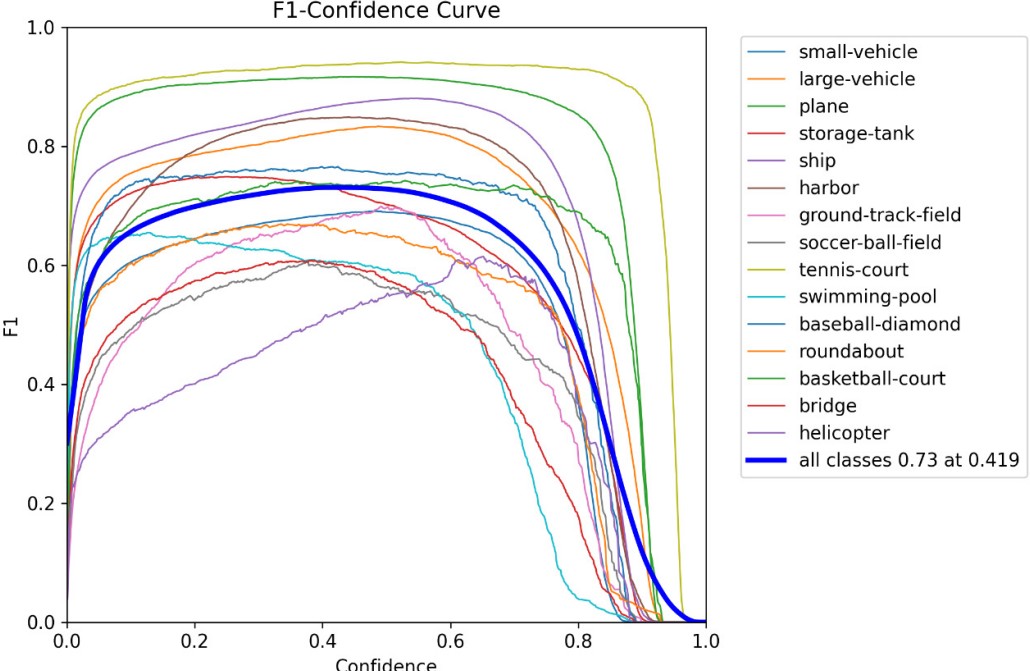

**Figure 13.** The F1 curve of the model.

Each row of the confusion matrix represents an actual category of the sample, as shown in Figure 14, while each column represents the predicted category outcome. The

entire matrix reflects the ability of the model to classify the targets in each category. The horizontal context represents FP, which is the negative sample share of prediction error. The vertical background represents FN, which is the positive sample share of prediction error. In the confusion matrix, the diagonal data represent the proportion of correctly classified categories. The categories that perform well in the confusion matrix are large vehicles, aircraft, ships, ports, and tennis courts with 0.84, 0.91, 0.87, 0.87, and 0.92, respectively, with football fields corresponding to the largest FN and significant leakage. The categories of swimming pools and bridges also corresponded to higher FNs. This is because the number of training samples for these categories is significantly lower than the number of training samples for the other categories. With only 1603 out of 208,365 instances, this makes it difficult to fit the model to the corresponding features, resulting in a poor feature extraction capability for the model to effectively identify such targets. On the other hand, although the accuracy of the correspondence between boats and small vehicles is high, the corresponding FP values are also high and there are some occurrences of false detection. Small vehicles are small or very small targets with stacking. In addition, the cutting of the images causes some small vehicles and boats in the dataset to be unmarked, which further affects their FP values.

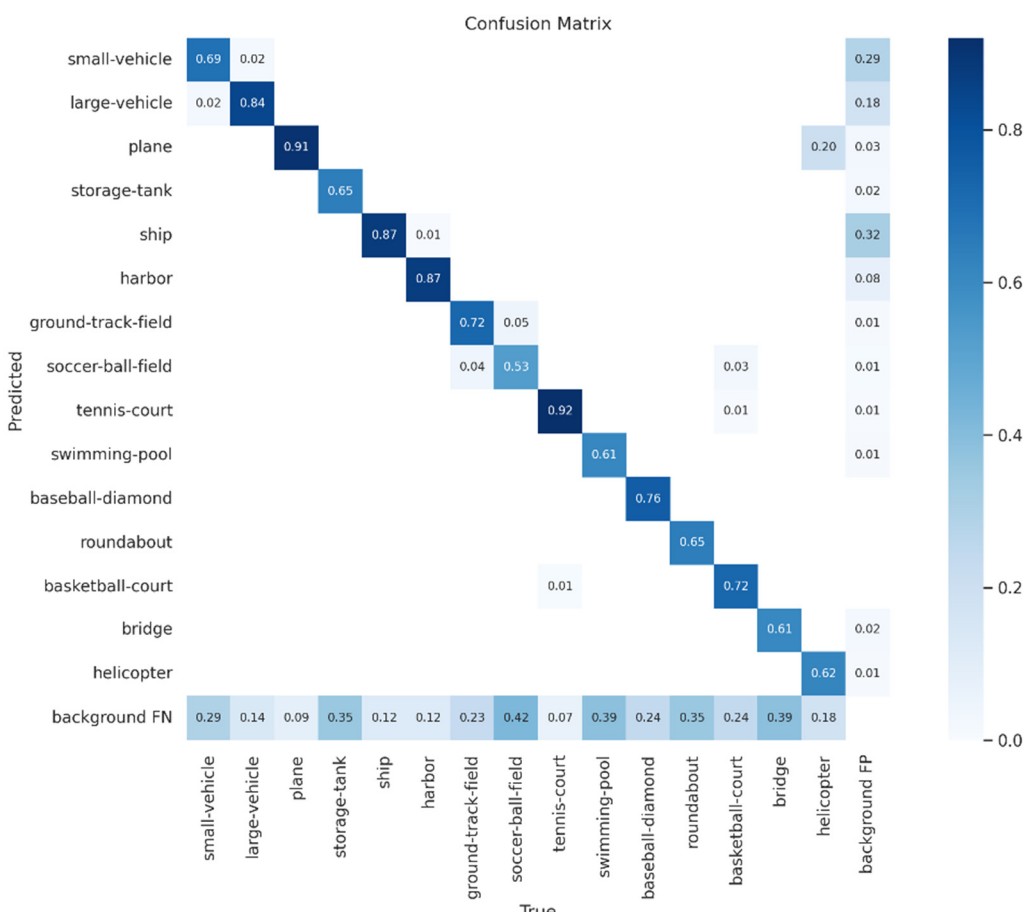

**Figure 14.** Confusion matrix.

Combined with the P-R curves of the improved YOLOv5 in Figure 11, the categories with fewer instances in the confusion matrix in the dataset include football fields and swimming pools which can be seen to have higher FN values, while according to the results before the improvement in the previous paper, the thesis optimized YOLOv5 model has 55.5% and 61% AP values in these two categories, respectively, compared to the original YOLOv5 algorithm before the improvement which had values of 50.3% and 53.3%, an improvement of 5.2% and 6.8%, respectively. This indicates that the improved algorithm

can achieve a high improvement in detecting categories with fewer instances in the dataset, is resistant to sample imbalance, and has a strong learning capability.

Also, for categories with high FP values in the confusion matrix, such as small cars and boats, the improved YOLOv5 algorithm achieves AP values of 53.2% and 79.3% for these two targets. The performance of the improved YOLOv5 algorithm is 69.6% and 88.7%, an improvement of 16.4% and 9.4%, respectively, which further improves the detection effect and reduces the occurrence of false detections in the two categories with high false detection rates. The improved algorithm has significantly improved the performance of the two categories of targets, which are small targets which are prone to stacking, indicating that the improved YOLOv5m algorithm can be well applied to the detection of small targets and target stacking.

### 3.3. Other Comparative Experiments

The DIOR dataset is also a large-scale benchmark dataset for target detection in optical remote sensing images. It contains 23,463 images in 20 categories and instances. There are 11,738 graphs in the test set, 5862 graphs in the training set, and 5863 graphs in the verification set. In this section, the model with the best improvement in Section 3.2 is selected to further validate the improved effectiveness on the DIOR dataset. The settings of the experiment-related hyperparameters remain the same as in the previous section. Table 7 shows the validation results for the DIOR dataset.

**Table 7.** Validation results for the DIOR dataset.

| Category | YOLOv5 | YOLOv5 for This Article |
|---|---|---|
| Aircraft | 79.9 | 95.1 |
| Airport | 73.3 | 87.9 |
| Baseball field | 72.1 | 94.9 |
| Basketball courts | 89.9 | 87.4 |
| Bridges | 46.4 | 60.3 |
| Chimneys | 80.7 | 92.0 |
| Service areas | 63.0 | 77.7 |
| Tollbooths | 66.6 | 77.8 |
| Dam | 48.9 | 75.8 |
| Golf course | 76.8 | 80.7 |
| Athletic fields | 73.5 | 84.4 |
| Ports | 57.2 | 70.7 |
| Highways | 58.8 | 70.9 |
| Boats | 91.1 | 95.0 |
| Stadiums | 60.3 | 93.3 |
| Storage Tanks | 79.6 | 86.4 |
| Tennis courts | 88.7 | 93.5 |
| Train station | 62.2 | 68.2 |
| Car | 61.5 | 81.9 |
| Windmill | 72.3 | 85.5 |
| mAP@0.5 | 70.0 | 82.9 |

It can be seen that the improved YOLOv5m model in this paper has different degrees of improvement in detection accuracy for all 20 categories on the DIOR remote sensing dataset compared to the original model, and that the mAP of the model is improved by 12.9%.

Figure 15 shows the convergence process of the improved YOLOv5m target detection network trained on the horizontally annotated DIOR dataset in this paper.

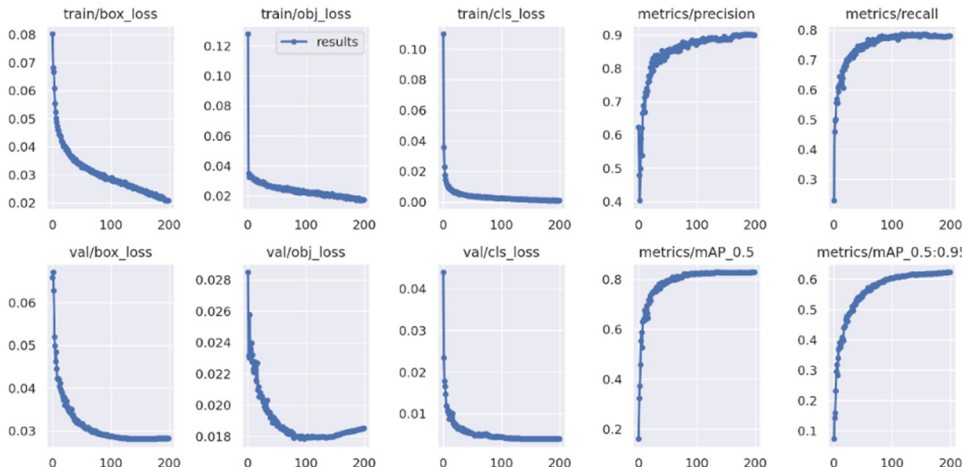

**Figure 15.** Network convergence on the DIOR dataset.

Figure 15 shows that when the network model is trained to 200 epochs, the regression and classification loss of the model on the validation set converge, and the mAP value of the model is also stable. This shows that the improved YOLOv5m model is stable and achieves good detection results on the horizontally annotated DIOR dataset, which is a significant improvement over the original model. Figure 16 shows the P–R curves of the improved YOLOv5m model trained on the horizontally annotated DIOR dataset.

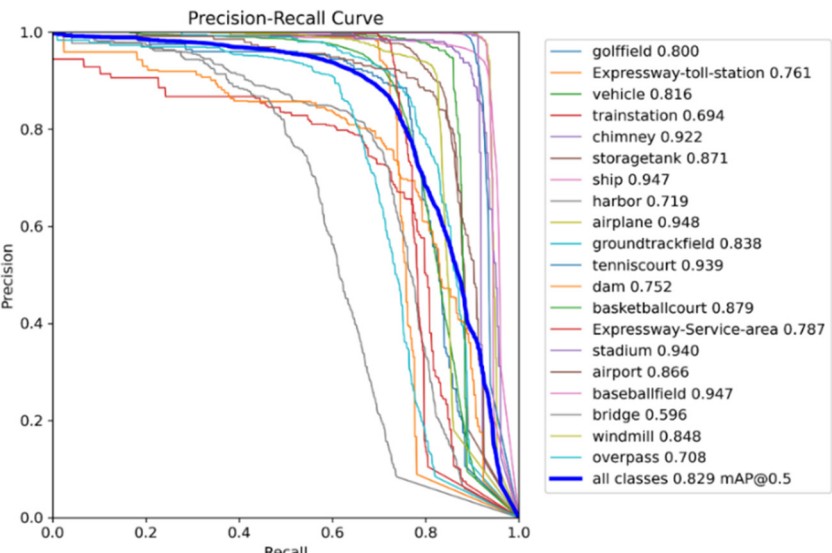

**Figure 16.** P–R curves on the DIOR dataset.

Observe that the area enclosed by each line of the P–R curve and the coordinates is the corresponding AP value for each category, with each curve being very close to the upper right corner of the coordinate axis. The area enclosed by the P–R curves for all categories is also large enough to be closer to the upper right corner of the coordinate graph. method, which also achieved good detection results on the DIOR dataset.

Figure 17 shows the detection results of the improved YOLOv5m target detection model on the horizontally annotated DIOR dataset. From the figure, it can be seen that the improved detection model also achieves good detection results on the DIOR dataset, with the detection accuracy of most categories of targets in the image being above 90%. Even for small targets such as vehicles and boats that only occupy a pixel level in the image, the detection accuracy can be around 80%.

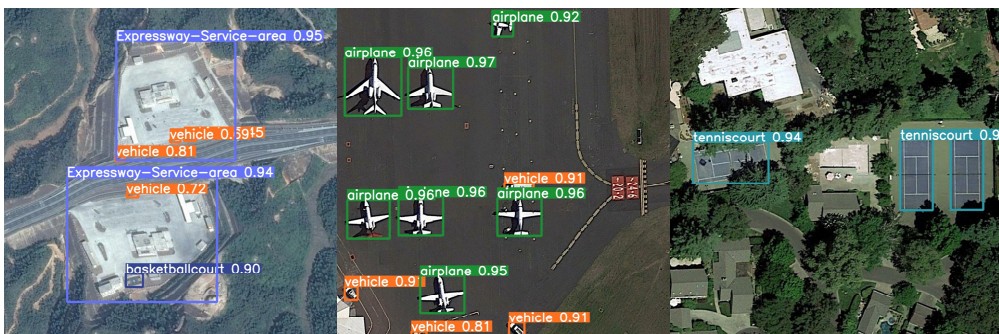

**Figure 17.** DIOR dataset detection results.

This summary validates the method previously proposed in this paper by comparing it to the DIOR dataset. The following section validates the effectiveness of the improved model by comparing it to the results achieved by the improved YOLOv5m algorithm on the horizontally annotated DOTA dataset.

Table 8 shows the performance of the improved YOLOv5 compared with several typical target detection models, such as RetinaNet, YOLOX, and the L model of YOLOv5, where Swin-YOLOv5 [19] and SPH-YOLOv5 [20] are the new v5 model structures proposed in 2022. As can be seen from Table 8, the improved YOLOv5 model in this paper achieves good detection results on the horizontally annotated DOTA dataset of remote sensing images, achieving results with the second mAP value.

**Table 8.** Comparison of results with other methods.

| Methods | mAP | Param (M) |
|---|---|---|
| RetinaNet | 0.675 | 36.7 |
| YOLOv5m | 0.642 | 21.2 |
| YOLOv5l | 0.740 | 46.5 |
| YOLOXm | 0.694 | 25.3 |
| Swin-YOLOv5 | 0.732 | - |
| SPH-YOLOv5 | 0.716 | - |
| **Improved YOLOv5m for this article** | **0.736** | **28.1** |

Although the detection effect is lower than that of the L model of YOLOv5, the mAP value is only 0.4% lower than that of the L model due to the fact that the number of parameters of the L model is 46.5 M, while the number of parameters of the improved YOLOv5 model in this paper is only 28.1 M. This is a 3% improvement over the other performance detector, YOLOX. In addition, when comparing the results of the other two improved YOLOv5 methods, the improved YOLOv5m method improves the mAP value by 0.4% and 2%, respectively, on the horizontally annotated DOTA dataset, compared to the two methods that introduced the Swin transformer and the self-attentive module. Compared to the pre-modified YOLOv5, a 9% improvement in mAP values was achieved with only a 6.9 M increase in the number of participants. This comparison with other methods further validates the validity and feasibility of this paper's improvement idea for the YOLOv5 model.

### 3.4. Comparison of Test Results

As shown in Figure 18, on the DIOR dataset, the same remote sensing image is used to verify and compare the detection effects of the original algorithm and the improved YOLOv5m method in this paper. The first column in the figure is the recognition result of YOLOv5, and the second column is the recognition effect of the improved YOLOv5m algorithm in this paper. The first column in the figure shows the recognition result of YOLOv5, and the second column shows the recognition effect of the improved YOLOv5m

algorithm in this paper. It can be seen that the improved YOLOv5 algorithm has a significant improvement in the effect of large target size span, small targets, and dense scenes. As before the improvement, small vehicles were not detected in Figure 18a and targets were not detected in the complex background scenario in Figure 18c. Also in Figure 18e, there was a case where a small car was mistakenly detected as an aircraft.

However, the small vehicles missed by the original algorithm in Figure 18b can be accurately identified and labelled. For the complex scene in Figure 18c, where there is a lot of background information and few targets, the improved algorithm filters out the background information to focus on the target information and identifies the targets that are not detected in Figure 18c in Figure 18d. For the denser scenario in Figure 18e, the original algorithm incorrectly detects large cars as aircraft. The improved algorithm successfully identifies the large vehicle correctly, while detecting other targets around it well. It can be seen that the attention mechanism and DDH algorithm can be well applied to remote sensing image target detection scenarios and has a great improvement effect on small targets with complex backgrounds and dense situations.

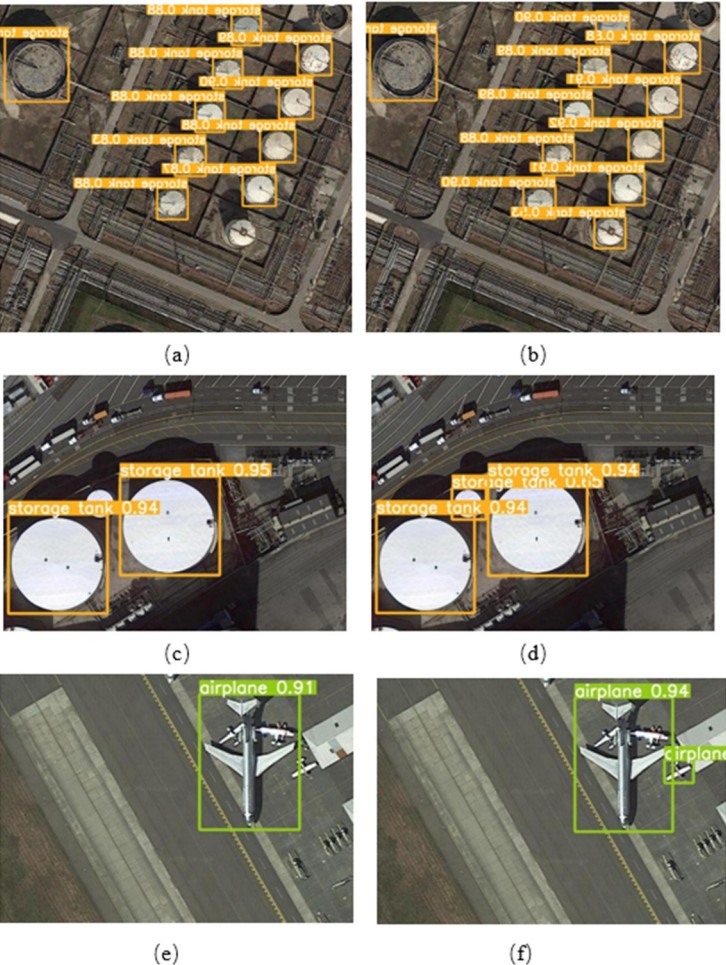

**Figure 18.** (**a**,**c**,**e**) shows the recognition results of YOLOv5, (**b**,**d**,**f**) shows the recognition effect of the improved YOLOv5m algorithm.

## 4. Conclusions

This paper focuses on the special problems of the remote sensing image scene, such as special perspective, multi-scale, multi-direction, small target, high background complexity, and so on. The YOLOv5 target detection algorithm is improved in the input stage, backbone network, and output stage, respectively.

In this paper, according to the characteristics of remote sensing images, the K-Means++ algorithm was used to improve the anchor generation method of the original YOLOv5 and apply it to the DOTA dataset. In addition, the image target template on the original coco dataset was replaced with the anchors template generated by K-Means++. The template has a recall rate of 0.9977 on the DOTA dataset and an increase of 0.6% on mAP.

The obtained template was applied to subsequent experiments. Firstly, the output end was improved by introducing DDH. At the same time, on the basis of separating the two types of tasks; IoU-aware was used to improve the correlation between classification accuracy and positioning accuracy. After many experiments on the DOTA dataset, it can be seen that the overall mAP of the improved output YOLOv5 is 3% to 4% higher than of the original model.

After introducing attention, the model is 5% to 9% higher than the original model on the mAP. Among them, the introduced CBAM attention mechanism has a better effect than the CA and SENet attention mechanisms, and the improvement effect reaches 9%. The improvement is more significant on small targets and dense targets.

After verification on the horizontally labelled DOTA dataset, the paper continues to compare the best improved model with the baseline model on the horizontally labelled DIOR remote sensing dataset. Compared with the original model, the improved method is improved by 12% on mAP. At the same time, this chapter compares the proposed method with other detection models, and achieves good detection results compared with Retina Net, YOLOX, Swin-YOLOv5, and other detection methods, which verifies the effectiveness of the proposed method.

In the future, we will conduct in-depth experiments on the DDH algorithm, adjust the hyperparameters, and study the most appropriate setting method to further improve the detection performance.

**Author Contributions:** Conceptualization, X.L. (Xiaodong Liu); Methodology, X.L. (Xiaodong Liu); Project administration, W.G.; Resources, Z.G.; Supervision, X.L. (Xiang Li); Validation, X.L. (Xiaodong Liu); Writing—original draft, L.S.; Writing—review and editing, L.S. All authors have read and agreed to the published version of the manuscript.

**Funding:** This research received no external funding.

**Data Availability Statement:** The data set used in the paper can be downloaded here: DOTA (captain-whu.github.io, accessed on 1 January 2023); https://aistudio.baidu.com/aistudio/datasetdetail/15179, accessed on 1 January 2023).

**Conflicts of Interest:** The authors declare no conflict of interest.

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
