# Peer review of "Remote Sensing Image Target Detection and Recognition Based on YOLOv5"

_remotesensing, doi:10.3390/rs15184459_

Round 1
Reviewer 1 Report
Dear editor,
I am glad to read this paper. However, i have some questions and suggestion for the authors to improve their manuscript.
The main contributions should be summarized.
In the introduction, why do you select the Yolov5m as the main primary network to improve?
For K-means++, CA, SE, and CBAM, these modules are used frequently in today's yolov5 improved models, so why were they chosen? What is the main difference?
Have you compared with other new modules proposed in last 1-2 years? not the yolov5 series.
Have you compared the models on other public datasets?
In your work, you focus on the small objects, how to show us the small object detection results?
The novelty should be summarized again.
Best wishes!
Reviewer 2 Report
This article aims to improve the performance of the yolov5 network with the help of several improvements.
The presented individual improvements in this article do not represent a novelty with regard to yolo network. Rather, it is an approach to modelling the network with different setups that include already existing impovements. Unfortunately, this objective was not adequately explained in the article.
The paper is very much designed for the Yolo community and less for someone who has not yet worked with the methodology. Some terms such as CBS or explanations of the procedures are missing (purpose of anchor boxes benchmarks?).
Neither k-means++, attention mechanisms nor decoupled detection represent a novelty for the community with regard to yolov5.
The improvements by K-Means++-were achieved for small objects and objects with a high frequency. From this it can be deduced that the improvement in overall accuracy was achieved by the classes with the higher frequency, which subsequently can be attributed to the imbalanced classes. The imbalanced problem in regard to classes and targets was not explicitly addressed in the article, although it is crucial for the F1 performance of individual classes.
The improvement by DDH is obvious and the improvement by attention mechanism is only partial. Why was the attention mechanism not investigated without DDH to gain additional inference?
The dataset contains 2806 aerial images, with the training set containing 1411 images and the validation set containing 458 images. Are the remaining images a test dataset? If so, indicate this. One cannot validate models on validation data alone. The models must be validated on an independent test data set.
Comparison with DIOR dataset:
What is the data setup for this experiment (train, val, test?)
There are no visual comparisons of the DOTA data set.
The wording in the article is basically okay, but there are passages whose content or use is not at all comprehensible, e.g. lines 225, 355 and 562 (sexual, mental, daily?).
There are some text formatting errors e.g. spacing between tables and text
Image captions are too short resp. were not displayed in a meaningful way. It is hard for the readers who do not belong to the Yolo community to understand the content. Figure 4.4 does not exist.
Figure 12 is superfluous as relevant information has already been presented in Figs. 8 and 9, or only the graph in the upper left corner of Figure 12 is sufficient.
Reviewer 3 Report
1.Kindly review the formatting throughout the manuscript, including text style, size, formulas, tables, figures, etc.
2.Minor revisions are needed to improve the style of the article, as it contains spelling and punctuation errors.
3.The authors describe the motivation and the problem well. But, during the description of the scientific background, the literature review is just a pile of information, lacking of analysis and induction.
4.We recommend including a dataset link to facilitate easy access for the scientific community.
5.We suggest adding a diagram showing the proposed work.
6.The authors should provide an explanation for their choice of hyperparameters in the proposed model, such as the number of epochs, batch size, etc. Moreover, why would one choose YOLOv5 over other versions?
7.Please provide more specific information about the results. Instead of simply stating that the proposed model achieved "the highest confidence, the best performance, and best result," include the exact accuracy score obtained in the experiments, along with other relevant metrics.
8.The article lacks a discussion on interpretability. Despite achieving high accuracy, the proposed method may not be easily understood by readers who seek insight into the factors influencing the classification decision. Sufficient discussion on enhancing the interpretability of the proposed method is needed.
Round 2
Reviewer 1 Report
Dear Editors,
I am glad to review this manuscript again, However, I still suggest the author considering and finding out the novelty again as these models and blocks are usually used in recent journals or papers.
Best wishes
Reviewer 2 Report
I hereby provide the feedback for authors after the revision:
Anchor box benchmarks: specify in the text the purpose of the benchmarks, e.g. dimensions for each pyramid feature map....
The problem of imbalance between classes and targets was not explicitly addressed in the article, although it is critical for F1 performance of each class: this was not explained in the article. Revise your problem statement or data definition to explain the imbalance problem, since you are using imbalanced data and thus the estimator may learn in favor of the classes with the majority of observations. In particular, the number of targets in each category is extremely imbalanced.
Comparison with DIOR dataset: What is the data setup for this experiment (train, val, test?); As you did not provide the answer, one can not understand if you use test data for this experimental stage. Indicate in the Figure 16 which data set is used. Indicate which data set is used in Chapter 3.4 Comparison of test results.
Major improvements to be done:
It is not clear in the article where you evaluate on the validation and where on the test data. You need to evaluate the generalizability of the model on the test data. Traininig performance and convergence may well be evaluated on the validation data - but that says nothing about the true model quality.
a) Calculate and show in a table the number of targets per class in the test data set of the DOTA data set.
One cannot see confusions in the Figure 14 Confusion matrix.
b) Therefore, calculate and show the correct and incorrect predictions per class in the test data by plotting the predictions for all targets per class in absolute numbers in a confusion matrix. The number of correct and incorrect target predictions per class must match the total number of targets per class in the test data.
This is the only way to derive confusions and see which classes collide with which classes.
You still have few typos in the text. Increase the spacing between tables and text.
